# Effects of Branched-Chain Amino Acids on Skeletal Muscle, Glycemic Control, and Neuropsychological Performance in Elderly Persons with Type 2 Diabetes Mellitus: An Exploratory Randomized Controlled Trial

**DOI:** 10.3390/nu14193917

**Published:** 2022-09-21

**Authors:** Takaaki Matsuda, Hiroaki Suzuki, Yoko Sugano, Yasuhiro Suzuki, Daisuke Yamanaka, Risa Araki, Naoya Yahagi, Motohiro Sekiya, Yasushi Kawakami, Yoshinori Osaki, Hitoshi Iwasaki, Koichi Hashimoto, Shin-Ichiro Takahashi, Yasushi Hada, Hitoshi Shimano

**Affiliations:** 1Department of Internal Medicine (Endocrinology and Metabolism), Faculty of Medicine, University of Tsukuba, Tsukuba 305-8575, Japan; 2Department of Rehabilitation Medicine, University of Tsukuba Hospital, Tsukuba 305-8576, Japan; 3Laboratory of Food and Physiological Models, Department of Veterinary Medical Science, Graduate School of Agricultural and Life Sciences, The University of Tokyo, Tokyo 133-8657, Japan; 4Department of Clinical and Translational Research Methodology, Faculty of Medicine, University of Tsukuba, Tsukuba 305-8575, Japan; 5Institute of Food Research, National Agriculture and Food Research Organization, Tsukuba 305-8642, Japan; 6Department of Laboratory Medicine, Faculty of Medicine, University of Tsukuba, Tsukuba 305-8575, Japan; 7Department of Animal Resource Sciences, Graduate School of Agricultural and Life Sciences, The University of Tokyo, Tokyo 113-8657, Japan; 8Department of Rehabilitation Medicine, Faculty of Medicine, University of Tsukuba, Tsukuba 305-8575, Japan

**Keywords:** branched-chain amino acids, soy protein, type 2 diabetes mellitus, skeletal muscle, insulin resistance, kynurenine, depressive symptoms

## Abstract

Although branched-chain amino acids (BCAA) are known to stimulate myofibrillar protein synthesis and affect insulin signaling and kynurenine metabolism (the latter being a metabolite of tryptophan associated with depression and dementia), the effects of BCAA supplementation on type 2 diabetes (T2D) are not clear. Therefore, a 24-week, prospective randomized open blinded-endpoint trial was conducted to evaluate the effects of supplementation of 8 g of BCAA or 7.5 g of soy protein on skeletal muscle and glycemic control as well as adverse events in elderly individuals with T2D. Thirty-six participants were randomly assigned to the BCAA group (*n* = 21) and the soy protein group (*n* = 15). Skeletal muscle mass and HbA1c, which were primary endpoints, did not change over time or differ between groups. However, knee extension muscle strength was significantly increased in the soy protein group and showed a tendency to increase in the BCAA group. Homeostasis model assessment for insulin resistance did not significantly change during the trial. Depressive symptoms were significantly improved in the BCAA group but the difference between groups was not significant. Results suggested that BCAA supplementation may not affect skeletal muscle mass and glycemic control and may improve depressive symptoms in elderly individuals with T2D.

## 1. Introduction

The prevalence of diabetes is increasing worldwide. The International Diabetes Federation estimated that the number of adults aged 20–79 years with diabetes will increase from 536.6 million in 2021 to 783.2 million in 2045, with the proportion of elderly individuals over 65 years of age with diabetes increasing during this period [1]. Elderly people have a risk of sarcopenia characterized by decreases in skeletal muscle mass, muscle strength, and physical performance [2]. Compared with persons without diabetes, those with type 2 diabetes (T2D) exhibited 20% lower knee extension strength [3], showed a faster decline in lower extremity muscle strength [4], and had a 1.5- to 2.0-fold greater risk of sarcopenia [5]. Diabetes accompanied by sarcopenia has been associated with deterioration of daily living activities [6], increased risk of falls [7], mortality [8], and poor glycemic control [9].

Resistance training (RT) and nutritional interventions are the two major non-pharmacological approaches to improve skeletal muscle mass and strength [10,11]. Recommended daily protein intake in elderly people is 1.0–1.2 g/kg/day [10,11], with higher recommendations for persons who are exercising or have acute or chronic illness [10]. However, about 30–50% of community-dwelling elderly people consumed less than the recommended amount of protein [12] as did the elderly with diabetes [13,14]. Moreover, it is difficult for elderly people to change dietary habits to achieve the recommended dietary protein intake.

Supplementation of protein or amino acids in beverages or powders for easy intake is one strategy to increase protein intake in the elderly. Since leucine stimulates protein synthesis through a mechanistic target of rapamycin complex 1 (mTORC1) [15], it is logical that supplementation of leucine, branched-chain amino acids (BCAA), or BCAA-rich protein may effectively increase skeletal muscle mass. In fact, a meta-analysis demonstrated that leucine supplementation significantly increased myofibrillar protein synthesis compared when with placebo [16,17].

A meta-analysis showed that BCAA-rich supplementation significantly increased muscle mass, muscle strength, and physical performance in elderly individuals with sarcopenia compared with a control diet [18]. However, few studies have assessed the effect of supplementation of protein or amino acids on skeletal muscle in elderly individuals with T2D [19,20,21].

Despite its effectiveness, there are several safety concerns regarding protein supplementation for persons with diabetes. A high-protein diet can cause glomerular hyperfiltration [22], leading to glomerular injury [23]. Since leucine upregulates mTORC1, subsequently inducing the suppression of insulin signal transduction by phosphorylation of insulin receptor substrate-1 (IRS-1) [24], leucine supplementation might worsen insulin resistance. An observational study showed that the serum concentrations of BCAA were positively correlated with a homeostasis model assessment of insulin resistance (HOMA-IR) [25]. On the other hand, an animal study showed that 24 weeks of supplementation with leucine in rats fed a high-fat diet improved insulin sensitivity [26], and BCAA supplementation in persons with chronic liver disease with insulin resistance reportedly improved HOMA-IR [27]. BCAA supplementation could also affect depressive symptoms. BCAAs compete with tryptophan, a precursor of serotonin associated with depression, for transport across the blood–brain barrier [28]. Leucine also inhibits kynurenine aminotransferase (KAT), which metabolizes tryptophan into kynurenine [29], which is associated with several diseases including depression, dementia, and cancer [30]. However, the effects of BCAA or other protein supplements on insulin resistance, renal function, depressive symptoms, or cognition have not been examined in persons with T2D. Moreover, intervention periods in most previous studies were within 3 months [18,31]. Longer intervention periods are needed to examine the effects of protein or amino acid supplementation on muscle mass and to reveal adverse effects.

Therefore, we conducted a 24-week randomized controlled study to investigate the effects of BCAA accompanied by standard exercise based on the American Diabetes Association’s recommendations [32] on skeletal muscle mass or strength, glucose metabolism, and neuropsychological performance in elderly persons with T2D and to evaluate the safety of BCAA supplementation.

## 2. Materials and Methods

### 2.1. Study Design and Participants

A 24-week, two-parallel-group, single-center, prospective randomized open, blinded-endpoint trial was performed at the University of Tsukuba Hospital from May 2019 to October 2020. The clinical research ethics committee of the University of Tsukuba Hospital approved the study protocol (H30-038, approved on 18 May 2018), which complies with the Declaration of Helsinki. This study was registered with the University Hospital Medical Information Network Clinical Trials Registry (UMIN000032368).

Participants were recruited from persons with T2D who regularly visited the outpatient clinic of University of Tsukuba Hospital. Inclusion criteria were: (i) age 65 to <80 years, (ii) HbA1c 6.5 to <8.5% at enrollment, and (iii) HbA1c change of ≤1.0% in the past six months. Exclusion criteria were: (i) diabetes other than T2D, (ii) receiving insulin, growth hormone, glucocorticoids, or anabolic steroids, (iii) estimated glomerular filtration rate (eGFR) < 30 mL/min/1.73 m^2^, (iv) having proliferative retinopathy, (v) inability to perform exercise training due to bone and joint disease, and (vi) undergoing treatment for malignancy. Every participant received a detailed explanation of the trial, and written informed consent was obtained from all participants before enrollment.

### 2.2. Intervention

Thirty-eight individuals met the inclusion criteria. Study participants were randomly assigned to the BCAA group or the soy protein group (1:1). Soy protein was used as the BCAA comparator to ensure that the differences in allocation would not cause differences in motivation to participate in the study and because the BCAA content in soy protein is lower than that in whey or casein [33]. The BCAA group consumed a BCAA-rich amino acid supplement (Delicious Amino Acids BCAA, Ezaki Glico Co., Ltd., Osaka, Japan) containing 8 g of BCAA (4 g leucine, 2 g valine, and 2 g isoleucine) and 36 kcal of energy. A study examining muscle protein synthesis (MPS) in elderly men showed that supplementation of 0.052 g/kg leucine, 0.0116 g/kg isoleucine, and 0.0068 g/kg valine significantly increased MPS when compared with controls [34]. In those with a body weight of 60 kg, this corresponds to a dose of 3 g leucine, 0.7 g isoleucine, and 0.4 g valine. Since each package of the supplement used contained 4 g of BCAA, two packets totaling 8 g of BCAA were administered. The soy protein group ingested a soy protein supplement (SAVAS Soy Protein 100, Meiji Co., Ltd., Tokyo, Japan) containing 7.5 g of soy protein and 40 kcal of energy. Although the amino acid content in the soy protein supplement was not disclosed by the manufacturer, the Ministry of Education, Culture, Sports, Science and Technology has published information regarding the amino acid content of isolated soy protein, denoting a content of 90 mg leucine, 52 mg isoleucine, 15 mg valine, and 436 mg essential amino acids per gram protein [33]. Those supplements were provided in individual packets for ease of use. All participants were instructed to take the supplements after exercise or after dinner if they did not exercise.

Participants were instructed to perform aerobic exercise and comprehensive resistance training at least three times a week. Comprehensive resistance training was to include stretching, 2 sets of 10 repetitions of thigh raises for each leg, 2 sets of 10 squats, 2 sets of 10 repetitions of seated leg lifts for each leg, and one-leg stands (30 s each leg).

A clinical research coordinator (CRC) monitored supplement intake once a week as well as compliance with the exercise program by e-mail or telephone to assure compliance with the intervention and to check for adverse effects.

Changes in anti-diabetic drugs were not allowed during the study period, unless the HbA1c level increased to ≥8.5% or decreased to <6.0% if the participants were taking sulfonylureas or glinides.

### 2.3. Outcomes

The primary outcomes of this study were changes in skeletal muscle mass and HbA1c level. Secondary outcomes included changes in muscle strength (knee extension muscle strength, knee extension muscle endurance, and grip strength), glucose metabolism (fasting plasma glucose [FPG], fasting serum insulin, and HOMA-IR), lipid metabolism (total cholesterol [TC], low-density lipoprotein cholesterol [LDL-C], high-density lipoprotein cholesterol [HDL-C], and triglycerides [TG]), neuropsychological performance (cognitive function evaluated by Mini Mental State Examination [MMSE]; depression and motivation assessed by the Quick Inventory of Depressive Symptomatology [QIDS] and Starkstein Apathy Scale [SAS], respectively), tryptophan metabolites (kynurenine and kynurenic acid), and renal function (estimated glomerular filtration rate [eGFR] and urinary albumin excretion [UAE]). In the post-hoc analysis, changes in insulin-like growth factor-1 (IGF-1), follistatin, and brain-derived neurotropic factor (BDNF) were evaluated.

### 2.4. Sample Size

The sample size was calculated based on a similar study with Japanese participants [35]. In that study, 6 g supplementation of essential amino acids including 42% leucine administered to elderly female participants increased the skeletal muscle mass by 0.3 ± 0.3 kg compared with placebo. Based on that experience, the minimum sample size was calculated as 16 per group when α = 0.05 and β = 0.20. Therefore, we set the recruitment goal as 20 participants per group considering the possibility of dropouts. However, recruitment of participants could not be carried out as planned due to the coronavirus disease 2019 (COVID-19) pandemic.

### 2.5. Randomization

A total of 38 participants were randomly assigned to the BCAA group or the soy protein group (1:1) based on the minimization method considering sex, age (<75 years or ≥75 years), and HbA1c level (<7.5% or ≥7.5%) at enrollment. Randomization was performed by CRC using electronic data capture. The allocation was blinded to attending doctors and examiners, though study participants were aware of their allocations through supplement packaging.

### 2.6. Physical Measurements

Body composition was measured using multi-frequency bioelectrical impedance analysis (BIA) (InBody 720, Biospace, Tokyo, Japan). All measurements were done with the participant in a fasting state and prior to muscle strength tests. Participants with an implanted cardiac pacemaker were excluded from the measurements. Body mass index (BMI) and skeletal muscle mass (SMI) were calculated by dividing body weight (kg) by the square of height (m^2^) and by dividing the limb skeletal muscle mass (kg) by the square of height (m^2^), respectively.

Grip strength of the dominant arm was measured using a Smedley analog grip strength meter (Toei Light, Saitama, Japan). Knee extension muscle strength and knee extension muscle endurance on the dominant foot side were measured using a torque machine (Biodex System 3, Sakai Medical, Tokyo, Japan). Knee extension muscle strength was measured at the maximum power in the isokinetic movement (60°/s). Three consecutive knee extension operations were performed, and the maximum value was adopted. Knee extension muscle endurance was evaluated by the total work of 20 continuous knee extensions with maximum effort in the isokinetic movement (300°/s).

### 2.7. Dietary Habits and Physical Activity

Participants completed a brief-type self-administered diet history questionnaire (BDHQ) [36] to estimate food intake at each visit with the assistance of the CRC. Daily physical activity was measured using an accelerometer (HJA750C, Omron, Japan). We used data from the accelerometer for days when it was worn at least 10 h based on another study’s data extraction method [37]. Moreover, we collected a self-administered questionnaire that surveyed time spent walking, performance of resistance training, and other activities.

### 2.8. Laboratory Analyses

Blood samples were collected in the morning after an overnight fast. A 24 h urine specimen was collected the day before the visit. Serum levels of urea nitrogen (UN), creatinine (Cre), TC, TG, and HDL-C, plasma glucose levels, and urinary levels of UN and Cre were measured using an automated analyzer (Hitachi High-Technologies, Tokyo, Japan). HbA1c was measured by high-performance liquid chromatography (TOSOH, Tokyo, Japan). Serum levels of LDL-C, insulin, and IGF-1 were measured using a homogeneous assay (Sekisui Medical, Tokyo, Japan), a fluorescence enzyme immunoassay (TOSOH, Tokyo, Japan), and an electro-chemiluminescence immunoassay (Roche Diagnostics, Tokyo, Japan), respectively. Follistatin and BDNF were measured using enzyme-linked immunosorbent assay kits (R&D Systems, Minneapolis, MN and FUJIFILM Wako Pure Chemical, Osaka, Japan, respectively). Biochemical assays were performed immediately after serum isolation, and the remaining serum was aliquoted and stored at −80 °C until assay.

Urinary albumin was measured using a turbidimetric immunoassay (Nittobo Medical, Tokyo, Japan). eGFR was calculated by an equation modified for the Japanese population: eGFR (mL/min/1.73 m^2^) = 194 × serum Cre^−1.0949^ × Age^−0.287^ × 0.739 (if female) [38]. The estimated dietary protein intake was calculated by Maroni’s formula: 6.25 × (0.031 × Body weight [kg] + daily extraction of urinary UN [g/day]) [39].

### 2.9. Serum Metabolite Analyses

Serum metabolite analyses were performed as previously described [40]. Briefly, 50 μL of serum samples were mixed with methanol to remove macromolecules, such as proteins. After centrifugation, 3 kDa-cutoff ultrafiltration, and lyophilization, the samples were dissolved in 200 μL of ultrapure water. These samples were then subjected to metabolome analysis by LC-MS/MS (LCMS-8030, Shimadzu, Kyoto, Japan) using the Method Package for Primary Metabolites ver. 2 (Shimadzu) according to the manufacturer’s protocol. The m/z values to measure amino acids and kynurenine concentrations were included in this method package, while the m/z values for kynurenic acid were determined in our laboratory using the MS/MS device (LCMS-8030). The m/z values used for kynurenic acid were 189.85 > 144.00.

### 2.10. Statistical Analysis

The distribution of data was checked using the Shapiro–Wilk test. Data were log-transformed if distribution was non-normal. Each result was shown as mean ± standard deviation or median (interquartile range). Based on distribution, continuous variables were compared using the unpaired *t*-test or the Mann–Whitney U test for two-group comparisons, or using a linear mixed model for repeated measures (MMRM) or the Wilcoxon signed-rank test for two-time-point comparisons. Fisher’s exact test was used for categorical variables. Changes from baseline to 8, 16, and 24 weeks were analyzed by an MMRM followed by Bonferroni correction for multiple comparisons. *p* < 0.05 was considered statistically significant. All analyses were performed by IBM SPSS Statistics 24 (Chicago, IL, USA).

## 3. Results

### 3.1. Recruitment and Baseline Characteristics of Participants

We recruited 48 patients with T2D; however, 10 did not have interest in participating in this study. The remaining 38 were randomly assigned to the BCAA group (*n* = 20) and the soy protein group (*n* = 18). Two participants in the soy protein group withdrew consent after randomization due to lack of confidence in continuation of the trial and a diagnosis of lung cancer, respectively. One participant with a history of cardiovascular disease in the BCAA group experienced a sudden cardiac death during the study, which was concluded to be unrelated to the intervention. Since that participant had completed the measurements until the 16th week, his data were included in the analysis. In addition, one participant assigned to the soy protein group mistakenly received the BCAA supplement and therefore was included in the BCAA group. In total, data on 21 participants in the BCAA group and 15 participants in the soy protein group were analyzed based on per-protocol analysis (Figure 1).

Table 1 shows the baseline characteristics of the participants. Baseline characteristics between the BCAA group and the soy protein group were not significantly different except for the number of participants taking glinides (0% in the BCAA group vs. 33% in the soy protein group, *p* = 0.008).

### 3.2. Dietary Intake and Daily Activities during the Intervention

Total energy intake, protein intake, and daily activity did not significantly change throughout the intervention (Table 2), although the group × time interaction indicated a significant difference between groups in the changes in daily activity.

### 3.3. Changes in Body Composition, Muscle Strength, and Glucose and Lipid Parameters during the Intervention

Throughout the intervention, skeletal muscle mass and HbA1c levels, which were primary endpoints of the current study, did not change over time or differ between groups (Table 3). However, knee extension muscle strength was significantly increased in the soy protein group (12 ± 19 Nm/kg, *p* = 0.023) and showed a trend toward increases in the BCAA group (10 ± 24 Nm/kg, *p* = 0.086). There was no significant difference between groups in knee extension muscle strength. In the BCAA group, knee extension muscle endurance was significantly decreased and grip strength was significantly increased after the 24-week intervention (Table 3). In the soy protein group, knee extension endurance and grip strength were not significantly changed after the intervention. However, there were no significant differences in changes in knee extension muscle endurance and grip strength between the BCAA group and the soy protein group (Table 3). FPG, fasting plasma insulin, and HOMA-IR did not significantly change over time in either group or did not differ between groups (Table 3). Although serum levels of TC, HDL-C, and TG did not change significantly after the intervention, serum LDL-C levels in the soy protein group were significantly decreased after the intervention (Table 3). Changes in serum LDL-C levels differed significantly between the BCAA and soy protein groups (0.0 ± 0.3 vs. −0.2 ± 0.3 mmol/L, *p* = 0.037).

### 3.4. Changes in Renal Function, Neuropsychological Performance, and Amino Acids and Kynurenine Levels during the Intervention

Although eGFR and UAE in the BCAA group and soy protein group were not significantly changed after the intervention, there were significant differences in changes in UAE between the BCAA and soy protein groups (Table 4). Depressive symptoms assessed by QIDS were significantly improved only in the BCAA group, but changes in QIDS did not differ significantly between groups. Serum levels of total amino acids, BCAA, essential amino acids, non-essential amino acids, large neutral amino acids (LNAA), leucine, valine, tryptophan, tyrosine, asparagine, and glutamine were significantly increased in the BCAA group. However, BCAA supplementation did not significantly alter serum levels of kynurenine and kynurenic acid (Table 4).

### 3.5. Changes in Serum Concentrations of Insulin-like Growth Factor 1, Follistatin, and Brain-Derived Neurotropic Factor during the Intervention

Serum levels of IGF-1, follistatin, and BDNF were measured to explore the mechanism of the increased muscle strength and improved depressive symptoms. No significant changes were observed in those three growth factors after the intervention (Table 5).

### 3.6. Adverse Events

The adverse events observed during the intervention period were as follows: sudden cardiac death (*n* = 1), constipation (*n* = 2), abdominal discomfort (*n* = 1), gastric cancer (*n* = 1), rib fracture (*n* = 1), prostate cancer (*n* = 1), colon polyps (*n* = 1), and heatstroke (*n* = 1) in the BCAA group and nausea (*n* = 1), upper respiratory infection (*n* = 1), rib fracture (*n* = 1), heatstroke (*n* = 1), and ureteral stone (*n* = 1) in the soy protein group. Events other than constipation, abdominal discomfort, and nausea were determined to be unrelated to the intervention.

## 4. Discussion

We conducted a randomized controlled trial to examine the effects of amino acid supplementation on skeletal muscle, glucose metabolism, and neuropsychological performance in elderly persons with T2D. This is the first report of a comprehensive evaluation of the effects of amino acid supplementation on renal function and cognitive function in addition to skeletal muscle mass and glycemic control in elderly persons with T2D. This study produced four major findings. First, both BCAA and soy protein supplementation increased muscle strength but not skeletal muscle mass, which was not associated with changes in serum levels of IGF-1, follistatin, and BDNF. Second, glycemic control and insulin resistance were not affected by either BCAA or soy protein supplementation. Third, BCAA supplementation significantly increased serum tryptophan levels and improved depressive symptoms. Finally, the supplementation of 8 g of BCAA or 7.5 g of soy protein was tolerable in elderly persons with T2D.

### 4.1. Relationship between Amino Acids Supplementation and Skeletal Muscle Mass and Strength

Although meta-analyses showed that protein or essential amino acid supplementation increased skeletal muscle mass or lean body mass in elderly persons [31,41], the effects of protein or amino acid supplementation on skeletal muscle mass and muscle strength were controversial in elderly persons with T2D [19,20,21]. Leenders et al. reported that administration of 7.5 g of leucine did not alter lean body mass, skeletal muscle mass, and muscle strength after a 6-month intervention [21]. Yamamoto et al. showed no significant changes in skeletal muscle mass and lean body mass, and significant changes in muscle strength, after a 48-week intervention with 6 g of an essential amino acid supplement containing 2.4 g of leucine [19]. Memelink et al. reported that a 21 g whey protein drink containing 3 g of leucine significantly increased skeletal muscle mass, lean body mass, and muscle strength [20]. In the current study, both the BCAA containing 4 g of leucine and 7.5 g of soy protein supplementation did not significantly increase skeletal muscle mass but increased muscle strength in elderly persons with T2D. A meta-analysis showed that the effects of protein or amino acid supplementation without concomitant nutritional therapy and exercise training on lean body mass and muscle strength were not significantly different compared with control groups [42]. On the other hand, protein supplementation combined with resistance training could increase lean body mass and muscle strength, with higher protein intake leading to greater increases in lean body mass in healthy non-obese elderly persons [43]. Regarding differences in amino acid supplements and protein supplements, it was reported that skeletal muscle’s mTORC1 activation by leucine was enhanced by BCAA and further enhanced by essential amino acids after resistance training [44]. Leenders et al. did not combine resistance training with supplementation [21]. We and Yamamoto et al. combined non-supervised resistance training with supplementation [19]. Memelink et al. employed a combination of supervised resistance training and high-intensity interval training with supplements [20]. Although resistance training increased skeletal muscle mass and/or strength in elderly persons with T2D [45], a systematic review showed that protein supplementation combined with resistance training was more effective than resistance training alone [41]. Furthermore, studies conducted by Yamamoto et al. and Memelink et al. showed that the increased skeletal muscle mass or strength in those combining amino acid or protein supplementation with resistance training was more pronounced than in those with resistance training alone [19,20]. Therefore, it is possible that the increases in muscle strength with BCAA or soy protein supplementation observed in this study were not solely due to resistance training. The mean daily protein intake other than from the protein supplements of our study participants, those of Leenders et al. [21], and those of Memelink et al. [20] were 1.0 g/kg/day, 1.0 g/kg/day, and 1.15 g/kg/day, respectively. Taken together, the inconsistent results shown in those studies, including ours, might be due to differences in the type of exercise training, the amount of daily protein intake, and the content of leucine and other essential amino acids in the protein supplements.

An unexpected finding in the current study was that BCAA supplementation significantly decreased knee extension muscle endurance despite the increase in knee extension muscle strength while soy protein supplementation did not significantly change knee extension muscle endurance after the intervention. To the best of our knowledge, that is a novel finding in terms of amino acid or protein supplementation and skeletal muscle endurance in elderly persons. Animal studies showed that supplementation of leucine induced slow-fiber-related genes and transformed skeletal muscle fiber type from fast-twitch to slow-twitch [46,47]. Furthermore, BCAA supplementation in healthy young adults during endurance exercise training significantly increased the expression of skeletal muscle peroxisome proliferator-activated receptor gamma coactivator-1a (PGC-1a) mRNA content after exercise, which is a marker of slow-twitch muscle fiber [48]. However, another study showed that leucine supplementation did not change the type of skeletal muscle fibers in healthy young men [49]. Since PGC-1a expression is decreased with age [50], it is possible that the decreased knee extension muscle endurance was due to decreased responsiveness of PGC-1a to BCAA. Since skeletal muscle endurance was assessed by isokinetic dynamometry, which is the gold standard for measuring thigh muscle strength with very high inter- and intra-rater reliability in the elderly [51], the discrepancy between knee extension muscle strength and knee extension muscle endurance was unlikely to be influenced by measurement errors.

Several growth factors or myokines can change muscle fiber type. Lynch et al. showed that IGF-1 administration increased the proportion of type IIB and type IIA fibers and decreased the proportion of type I fibers in dystrophic mice [52]. Barbe et al. reported that overexpression of follistatin in mice increased the myosin isoform related to fast type IIB fibers and decreased the myosin isoform related to intermediate fast-oxidative types IIA and IIX fibers or slow type I fibers [53]. Delezie et al. also reported that the muscle-specific ablation or overexpression of BDNF was associated with the composition of muscle fiber types [54]. However, in the current study, there were no significant changes in those growth factors after the intervention. The cause of the divergent effects of BCAA or soy protein supplementation on knee extension strength and endurance remains to be elucidated.

### 4.2. Relationship between Amino Acid Supplementation and Glucose and Lipid Metabolism

Studies assessed by hyperinsulinemic-euglycemic clamp showed that amino acid infusion or acute ingestion of leucine or protein deteriorated muscle insulin resistance [55,56]. Conversely, a meta-analysis showed that whey protein administration significantly reduced HbA1c, insulin, TG, and LDL-C levels [57]. Leenders et al. reported that changes in FPG, HbA1c, fasting insulin, HOMA-IR, TG, TC, HDL-C, and LDL-C levels after a 24-week leucine supplementation did not differ significantly compared with placebo [21]. Yamamoto et al. showed that a 48-week leucine-rich essential amino acid supplementation did not significantly change HbA1c levels [19]. On the other hand, Memelink et al. reported that a leucine-enriched whey protein drink significantly improved insulin resistance assessed by HOMA-IR and the Matsuda Index [58] compared with placebo [20]. In the current study, FPG, serum insulin, HOMA-IR, HbA1c, TC, HDL-C, and TG levels were not significantly changed in either group during the intervention. Since skeletal muscle mass has a pivotal role in insulin sensitivity [59], the difference in the results between those of Memelink et al. and other groups, including ours, might not be due to direct effects of protein supplementation on insulin resistance but rather due to an indirect effect via increased skeletal muscle mass. The reduction in LDL-C levels in the soy protein group in the current study may have been due to soy-isoflavones and bioactive peptides in soy protein [60].

### 4.3. Relationship between BCAA Supplementation and Neuropsychological Performance

An observational study showed that circulating BCAA levels were lower in persons with major depression compared with healthy participants and were negatively correlated with the Hamilton Depression Rating Scale (HAMD-17) and Beck Depression Inventory (BDI) scores [61]. A cross-sectional study showed that groups with higher dietary BCAA intakes had lower odds of depression and anxiety in comparison with the lowest-intake group [62]. Moreover, Walker et al. reported that supplementation of L-leucine (50 mg/kg) inhibited lipopolysaccharide-induced depression-like behavior in mice [63]. In the current study, QIDS scores were significantly improved in the BCAA group. Taken together, BCAA supplementation could improve depressive symptoms. This is the first intervention study to demonstrate the beneficial effects of BCAA on depressive symptoms.

We considered three possibilities for the mechanism of improvement in depressive symptoms with BCAA supplementation. First, it was reported that high circulating levels of kynurenine or a high kynurenine-to-tryptophan ratio were associated with an increased risk of depressive symptoms [64,65]. Jonsson et al. reported that a single administration of BCAA during exercise decreased plasma levels of kynurenine but did not change plasma levels of tryptophan [48]. Although serum kynurenine levels and the kynurenine-to-tryptophan ratio in our study did not change during measurements in the fasting state, those levels might transiently decrease after ingestion of BCAA, subsequently improving the depressive state in the BCAA group. Second, the uptake of kynurenine at large amino transporter-1 (LAT-1) in the blood–brain barrier might be decreased by competing with an LNAA including BCAA, tryptophan, methionine, histidine, tyrosine, and phenylalanine [63,66]. The increasing serum LNAA values in our study support this possible mechanism. Third, the increased glutamine level might affect its uptake in the brain. Those increases may be due to the BCAA–branched-chain keto acid cycle [67]. Glutamine or glutamic acid values in the anterior cingulate cortex of persons with major depressive disorders were shown to be lower than in healthy control study participants [68]. Baek et al. reported that supplemental glutamine increased glutamine transporters in the prefrontal cortex of chronic-immobilization-stressed mice [69]. Taken together, the increased glutamine levels in the BCAA group might lead to improvements in the depressive state via increasing the concentration of glutamine or glutamic acid in the brain.

### 4.4. Protein Supplementation and Renal Function

Dietary protein restriction slows the decline in eGFR and reduces the risk of progression to end-stage renal disease in persons with chronic kidney disease [70]. The Kidney Disease: Improving Global Outcomes guidelines recommends 0.8 protein g/kg/day in persons with diabetes and chronic kidney disease (CKD) [71]. However, a protein-restricted diet for CKD conflicts with the high-protein diet recommended for prevention of sarcopenia. Observational studies showed that higher intakes of plant-source protein were significantly associated with slower declines in eGFR in elderly women with CKD [72,73]. In addition, supplementation of 7.5 g/day leucine for 6 months in elderly male persons with T2D did not change creatinine clearance [21]. In the current study, changes in eGFR and UAE did not differ significantly after the intervention in both the BCAA group and the soy protein group. Therefore, both 8 g of BCAA supplementation and 7.5 g of soy protein supplementation are highly likely to be safe with regard to renal function in elderly persons with T2D.

### 4.5. Strengths and Limitations

This study has two strengths. Changes in blood amino acid fractions and kynurenine concentrations with administration of BCAA or soy protein were analyzed. Previously there have been no reports evaluating changes in amino acid fractions and some kynurenine metabolites due to their supplementation. Second, this is the first report of a comprehensive evaluation of the effects of amino acid supplementation on renal function and cognitive function. These studies provide insights into the relationship between amino acid metabolism and neuropsychological performance.

This study has five limitations. First, it was a pilot study and did not have a control group. As previously mentioned, soy protein was used as a comparator to prevent loss of motivation for participation in the control group. In the future, randomized controlled trials with an appropriate sample size based on results of this study and with a control group comprising a non-protein-supplemented group or an exercise-only group should be conducted. Second, it is possible that the COVID-19 pandemic affected the activity of participants. Although there was no significant difference in daily activities as assessed by accelerometer between the groups, there was a significant group × time interaction. Third, body composition was assessed using BIA rather than dual-energy x-ray absorptiometry (DXA). Although appendicular skeletal muscle mass assessed by BIA was reportedly well-correlated with that assessed by DXA in Japanese elderly persons [74], the accuracy of BIA can be influenced by consumption of foods and water, exercise, and skin temperature; in addition, changes in body-fat mass can be ≥1% [75,76]. In the current study, the timing of the examination was constant, with morning measurements after an overnight fast. Fourth, since resistance training was not performed under supervision, it cannot be ruled out that variations in compliance could have affected the results. Finally, since our study groups were composed of those with both mild depressive symptoms and normal cognitive function, the effects of BCAA on depression and cognitive function may have been underestimated. To clarify the effects of BCAA on depression, more homogenous groupings are needed. Randomized controlled studies with depression as the primary endpoint are needed to confirm the beneficial effect of BCAA supplementation on depressive symptoms.

## 5. Conclusions

The present study describes the beneficial effects of amino acid supplementation on muscle strength in the elderly with T2D and demonstrated its safety regarding insulin resistance and renal function. As to the effect on skeletal muscle, supplementation of BCAA did not show superiority over soy protein supplementation. However, it was suggested that BCAA supplementation improves depressive states and soy protein supplementation decreases serum low-density lipoprotein concentrations. Those findings could be valuable for improving muscle function and managing comorbidities in elderly individuals with T2D.

## Figures and Tables

**Figure 1 nutrients-14-03917-f001:**
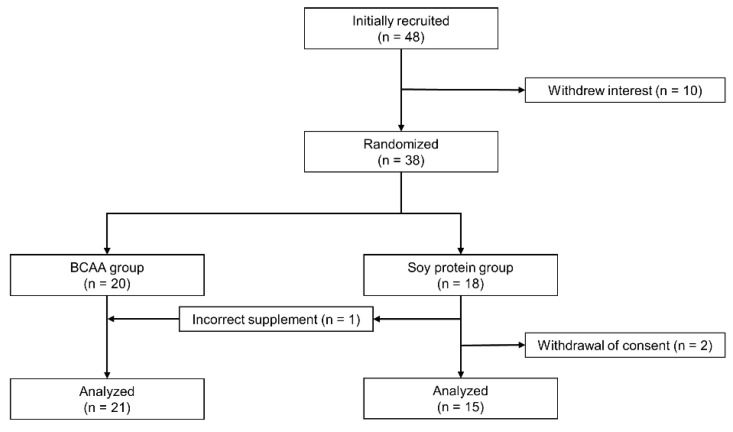
Flow chart of study participants. Participants (*n* = 38) were randomly assigned to the branched-chain amino acids (BCAA) group and the soy protein group based on minimization methods considering sex, age, and baseline HbA1c. Participants who were assigned to the soy protein group at first received BCAA supplement by accident and were finally assigned to the BCAA group.

**Table 1 nutrients-14-03917-t001:** Baseline clinical characteristics of study participants.

	BCAA	Soy Protein	*p*
*n*	21	15	
Age (years)	73 ± 4	73 ± 4	0.883
Female (%)	38	33	1.000
Duration of diabetes (years) *	21 (16–24)	19 (14–29)	0.431
Current smoking (%)	5	7	1.000
Alcohol consumption (g/day)	0 (0–10)	0 (0–0)	0.526
Total energy (kcal/day)	2017 ± 488	1804 ± 545	0.226
Protein intake (g/kg/day) *	1.0 (0.8–1.3)	1.0 (0.9–1.4)	0.228
Body mass index (kg/m^2^) *	24.3 (22.5–26.0)	23.1 (20.9–24.6)	0.132
Skeletal muscle mass (kg)	24.8 ± 5.4	23.4 ± 4.6	0.437
Skeletal muscle index (kg/m^2^)	7.0 ± 1.2	6.6 ± 0.9	0.259
Lean body mass (kg) *	45.9 (36.0–55.0)	44.1 (36.7–50.4)	0.520
Body fat percentage (%)	28.4 ± 8.0	25.9 ± 6.4	0.339
Grip strength (kg)	32 ± 10	31 ± 8	0.882
Knee extension strength (Nm/kg)	140 ± 40	146 ± 50	0.694
Knee extension endurance (J)	844 ± 287	762 ± 286	0.404
MMSE (points)	28 (27–30)	29 (27–30)	0.590
QIDS (points)	6 (3–8)	4 (2–6)	0.191
SAS (points)	11 (4–16)	13 (4–15)	0.751
Fasting plasma glucose (mmol/L)	7.6 ± 1.5	8.1 ± 1.2	0.324
Fasting serum insulin (pmol/L) *	29 (17–40)	20 (13–40)	0.463
HOMA-IR	1.4 (1.0–2.0)	1.2 (0.7–2.3)	0.590
HbA1c (mmol/mol)	56 ± 7	53 ± 3	0.190
Total cholesterol (mmol/L)	4.4 ± 0.8	4.7 ± 0.5	0.353
LDL-C (mmol/L)	2.4 ± 0.7	2.6 ± 0.4	0.293
HDL-C (mmol/L)	1.5 ± 0.4	1.6 ± 0.4	0.532
Triglycerides (mmol/L)	1.1 ± 0.5	1.0 ± 0.4	0.459
eGFR (mL/min/1.73 m^2^)	69 ± 16	66 ± 16	0.524
Urinary albumin excretion (mg/day)	23 (5–94)	13 (6–51)	0.547
Diabetic Complications			
Neuropathy (%)	24	13	0.674
Retinopathy (%)	57	60	1.000
Nephropathy (%)	38	33	1.000
Cardiovascular disease (%)	24	33	0.709
Antidiabetic drugs			
Metformin (%)	52	67	0.501
Sulfonylureas (%)	29	13	0.424
Glinides (%)	0	33	0.008
Thiazolidines (%)	5	7	1.000
SGLT2 inhibitors (%)	38	13	0.142
DPP-4 inhibitors (%)	62	73	0.721
GLP-1 receptor agonists (%)	14	7	0.626
α-glucosidase inhibitors (%)	24	27	1.000

Data are mean ± SD, median (interquartile range), or percentage. * log-transformed variables were used for the analyses. DPP-4, dipeptidyl peptidase-4; eGFR, estimated glomerular filtration rate; GLP-1, glucagon like peptide-1; HDL-C, high-density lipoprotein cholesterol; HOMA-IR, homeostasis model assessment of insulin resistance; LDL-C, low-density lipoprotein cholesterol; MMSE, Mini Mental State Examination; QIDS, Quick Inventory of Depressive Symptomatology; SAS, Starkstein Apathy Scale; SGLT2, sodium-glucose cotransporter 2.

**Table 2 nutrients-14-03917-t002:** Dietary intake and activity during the 24-week intervention period.

	Group	Baseline	8 Weeks	16 Weeks	24 Weeks	*p* for Time	*p* for Groups	*p* for Group × Time
Total energy (kcal/day)	B	2017 ± 488	1890 ± 555	1897 ± 463	1997 ± 645	0.623	0.112	0.958
	S	1804 ± 545	1670 ± 556	1599 ± 499	1712 ± 533	0.476		
Protein intake (g/kg/day) *	B	1.0 (0.8–1.3)	1.0 (0.8–1.4)	1.1 (0.8–1.3)	1.0 (0.8–1.3)	0.943	0.123	0.931
	S	1.0 (0.9–1.4)	1.2 (1.0–1.6)	1.2 (1.0–1.4)	1.1 (1.0–1.3)	0.744		
Daily activity (kcal/day)	B	654 ± 161	633 ± 142	643 ± 137	613 ± 130	0.282	0.430	0.027
	S	605 ± 189	590 ± 186	595 ± 191	636 ± 219	0.128		

Data are mean ± SD or median (interquartile range). * log-transformed variables were used for the analyses. B, branched-chain amino acid (BCAA) group; S, Soy protein group.

**Table 3 nutrients-14-03917-t003:** Changes in body composition, muscle strength, and glucose and lipid parameters during the 24-week intervention period.

	Group	Baseline	8 Weeks	16 Weeks	24 Weeks	*p* for Time	Changes between Baseline and 24 Weeks	*p* for Groups
Body mass index (kg/m^2^) *	B	24.3 (22.5–26.0)	24.1 (22.4–26.0)	23.7 (22.4–25.7)	24.0 (22.4–24.8)	0.396	−0.2 (−0.3–0.2)	0.222
	S	23.1 (20.9–24.6)	23.2 (21.0–24.6)	23.2 (20.8–24.6)	22.9 (20.6–24.6)	0.116	0.1 (−0.1–0.2)	
Skeletal muscle mass (kg)	B	24.8 ± 5.4	24.7 ± 5.5	24.8 ± 5.6	24.3 ± 5.2	0.940	−0.1 ± 0.8	0.693
	S	23.4 ± 4.6	23.7 ± 4.6	23.5 ± 4.5	23.4 ± 4.5	0.444	0.0 ± 0.5	
Knee extension strength (Nm/kg)	B	140 ± 40	—	—	148 ± 40	0.086	10 ± 24	0.781
	S	146 ± 50	—	—	159 ± 41	0.023	12 ± 19	
Knee extension endurance (J)	B	844 ± 287	—	—	780 ± 263	0.036	−44 ± 90	0.526
	S	762 ± 286	—	—	744 ± 205	0.645	−18 ± 149	
Grip strength (kg)	B	32 ± 10	—	—	33 ± 11	0.044	2 ± 3	0.213
	S	31 ± 8	—	—	32 ± 9	0.486	0 ± 2	
HbA1c (mmol/L)	B	56 ± 7	55 ± 6	56 ± 6	55 ± 5	0.711	−1 ± 4	0.140
	S	53 ± 3	55 ± 5	55 ± 7	55 ± 8	0.386	2 ± 6	
Fasting plasma glucose (mmol/L)	B	7.6 ± 1.5	7.7 ± 1.7	7.6 ± 1.2	7.8 ± 1.1	0.864	0.1 ± 0.7	0.780
	S	8.1 ± 1.2	8.9 ± 2.4	7.9 ± 1.1	8.1 ± 1.2	0.156	0.0 ± 1.0	
Fasting serum insulin (pmol/L) *	B	29 (17–40)	29 (15–41)	28 (19–41)	26 (16–33)	0.711	0 (−8–6)	0.182
	S	20 (13–40)	20 (14–38)	30 (17–34)	20 (10–28)	0.315	−1 (−11–2)	
HOMA-IR *	B	1.4 (1.0–2.0)	1.4 (0.9–2.5)	1.6 (0.9–2.3)	1.5 (0.9–1.9)	0.762	0.0 (−0.3–0.4)	0.153
	S	1.2 (0.7–2.3)	1.2 (0.8–2.3)	1.5 (0.8–2.1)	1.3 (0.6–1.6)	0.320	−0.1 (−0.7–0.2)	
Total cholesterol (mmol/L)	B	4.4 ± 0.8	4.4 ± 0.9	4.5 ± 0.9	4.5 ± 0.8	0.648	0.0 ± 0.3	0.145
	S	4.7 ± 0.5	4.6 ± 0.6	4.5 ± 0.6	4.5 ± 0.4	0.268	−0.2 ± 0.3	
LDL-C (mmol/L)	B	2.4 ± 0.7	2.3 ± 0.7	2.4 ± 0.7	2.5 ± 0.7	0.481	0.0 ± 0.3	0.037
	S	2.6 ± 0.4	2.5 ± 0.5	2.5 ± 0.5	2.4 ± 0.4 ^§^	0.044	−0.2 ± 0.3	
HDL-C (mmol/L)	B	1.5 ± 0.4	1.4 ± 0.5	1.4 ± 0.4	1.5 ± 0.4	0.602	0.0 ± 0.1	0.565
	S	1.6 ± 0.4	1.5 ± 0.4	1.5 ± 0.4	1.5 ± 0.4	0.177	0.0 ± 0.2	
Triglycerides (mmol/L)	B	1.1 ± 0.5	1.1 ± 0.5	1.1 ± 0.5	1.0 ± 0.5	0.372	−0.1 ± 0.3	0.801
	S	1.0 ± 0.4	1.0 ± 0.3	1.0 ± 0.3	0.9 ± 0.4	0.934	0.0 ± 0.4	

Data are mean ± SD or median (interquartile range). * log-transformed variables were used for the analyses. ^§^
*p* < 0.05 vs. Baseline. B, branched-chain amino acids (BCAA) group; HDL-C, high-density lipoprotein cholesterol; HOMA-IR, homeostasis model assessment of insulin resistance; LDL-C, low-density lipoprotein cholesterol; S, Soy protein group.

**Table 4 nutrients-14-03917-t004:** Changes in renal function, neuropsychological performance, and amino acid and kynurenine metabolism during the intervention.

	Group	Baseline	24 Weeks	*p* for Time	Changes	*p* for Groups
eGFR (mL/min/1.73 m^2^)	B	69 ± 16	71 ± 14	0.469	0 ± 6	0.241
	S	66 ± 16	68 ± 17	0.602	2 ± 7	
Urinary albumin excretion (mg/day) *	B	23 (5–94)	11 (4–52)	0.258	−3 (−23–1)	0.047
	S	13 (6–51)	12 (4–53)	0.386	−1 (−2–23)	
MMSE (points)	B	28 (27–30)	29 (26–30)	0.667	0 (−1–1)	0.086
	S	29 (27–30)	30 (28–30)	0.055	1 (0–1)	
QIDS (points)	B	6 (3–8)	4 (2–7)	0.019	−2 (−4–1)	0.400
	S	4 (2–6)	3 (2–4)	0.149	−1 (−3–1)	
SAS (points)	B	11 (4–16)	10 (4–16)	0.599	−1 (−3–2)	0.298
	S	13 (4–15)	12 (8–16)	0.461	1 (−2–4)	
Total amino acids (μmol/L)	B	3004 ± 261	3149 ± 198	0.010	145 ± 226	0.382
	S	2884 ± 267	2960 ± 202	0.232	76 ± 235	
Branched-chain amino acids (μmol/L)	B	407 ± 75	446 ± 68	0.015	39 ± 66	0.381
	S	391 ± 62	412 ± 76	0.186	21 ± 57	
Essential amino acids (μmol/L)	B	840 ± 111	891 ± 87	0.034	51 ± 99	0.629
	S	812 ± 91	847 ± 99	0.175	35 ± 94	
Non-essential amino acids (μmol/L)	B	2154 ± 189	2250 ± 155	0.021	96 ± 171	0.348
	S	2063 ± 200	2104 ± 127	0.365	41 ± 169	
Large neutral amino acids (μmol/L)	B	783 ± 105	836 ± 82	0.025	53 ± 97	0.527
	S	764 ± 84	796 ± 96	0.170	33 ± 87	
Leucine (μmol/L)	B	124 ± 26	135 ± 23	0.014	12 ± 19	0.224
	S	121 ± 23	125 ± 24	0.601	3 ± 22	
Isoleucine (μmol/L)	B	74 ± 15	78 ± 16	0.184	4 ± 13	0.650
	S	71 ± 14	77 ± 18	0.075	6 ± 12	
Valine (μmol/L)	B	209 ± 36	233 ± 32	0.012	24 ± 38	0.303
	S	198 ± 28	210 ± 36	0.109	12 ± 26	
Tryptophan (μmol/L)	B	50 ± 8	55 ± 9	0.004	5 ± 6	0.254
	S	50 ± 10	52 ± 7	0.329	2 ± 7	
Lysine (μmol/L) *	B	115 (106–126)	121 (109–129)	0.198	2 (−7–12)	0.854
	S	107 (93–120)	112 (101–124)	0.225	6 (−7–22)	
Methionine (μmol/L)	B	25 ± 4	27 ± 4	0.080	2 ± 4	0.393
	S	23 ± 3	24 ± 4	0.696	0 ± 5	
Phenylalanine (μmol/L)	B	58 ± 10	61 ± 9	0.054	3 ± 7	0.499
	S	55 ± 10	56 ± 10	0.622	1 ± 10	
Threonine (μmol/L) *	B	127 (119–139)	122 (113–135)	0.594	−2 (−18–10)	0.366
	S	130 (116–148)	142 (118–154)	0.264	6 (−10–17)	
Histidine (μmol/L)	B	53 ± 9	53 ± 7	0.919	0 ± 7	0.485
	S	51 ± 7	52 ± 7	0.435	2 ± 7	
Tyrosine (μmol/L)	B	61 ± 10	66 ± 13	0.028	5 ± 9	0.297
	S	60 ± 9	62 ± 9	0.340	2 ± 7	
Aspartic acid (μmol/L)	B	37 ± 7	38 ± 7	0.645	1 ± 10	0.757
	S	36 ± 5	36 ± 7	0.979	0 ± 8	
Asparagine (μmol/L)	B	73 ± 7	77 ± 9	0.042	4 ± 8	0.766
	S	71 ± 8	74 ± 9	0.212	3 ± 9	
Serine (μmol/L)	B	195 ± 35	203 ± 35	0.183	7 ± 23	0.457
	S	191 ± 21	192 ± 19	0.864	1 ± 24	
Glutamic acid (μmol/L)	B	53 ± 14	54 ± 11	0.723	1 ± 15	0.225
	S	53 ± 15	47 ± 17	0.217	−5 ± 16	
Glutamine (μmol/L)	B	1062 ± 110	1115 ± 98	0.022	53 ± 95	0.534
	S	986 ± 122	1018 ± 86	0.250	32 ± 104	
Proline (μmol/L) *	B	82 (73–93)	87 (78–105)	0.144	2 (−7–11)	0.068
	S	88 (75–104)	78 (74–93)	0.126	−4 (−13–4)	
Glycine (μmol/L)	B	209 ± 28	207 ± 29	0.761	−2 ± 29	0.398
	S	203 ± 33	209 ± 33	0.384	6 ± 26	
Alanine (μmol/L)	B	300 ± 47	317 ± 43	0.085	17 ± 42	0.249
	S	295 ± 62	297 ± 67	0.766	2 ± 29	
Arginine (μmol/L)	B	78 ± 16	82 ± 18	0.141	4 ± 13	0.832
	S	79 ± 18	82 ± 16	0.267	3 ± 12	
Cystine (μmol/L) *	B	9 (6–11)	6 (4–9)	0.137	−3 (−6–2)	0.772
	S	8 (4–12)	7 (5–12)	0.872	3 (−4–7)	
Kynurenine (μmol/L)	B	4.4 ± 0.9	4.5 ± 0.8	0.925	0.0 ± 1.0	0.521
	S	4.2 ± 0.9	4.4 ± 0.9	0.256	0.2 ± 0.7	
Kynurenic acid (μmol/L) *	B	0.04 (0.03–0.04)	0.03 (0.03–0.04)	0.449	0.00 (−0.01–0.01)	0.218
	S	0.04 (0.03–0.05)	0.04 (0.03–0.06)	0.587	0.01 (−0.01–0.01)	
Tryptophan to large neutral amino acid ratio *	B	0.06 (0.06–0.07)	0.07 (0.06–0.07)	0.313	0.00 (0.00–0.01)	0.641
	S	0.07 (0.06–0.07)	0.07 (0.06–0.07)	0.955	0.00 (0.00–0.00)	
Tryptophan to branched-chain amino acids ratio *	B	0.12 (0.11–0.14)	0.13 (0.11–0.14)	0.823	0.00 (−0.01–0.01)	0.783
	S	0.13 (0.12–0.15)	0.12 (0.12–0.15)	0.865	0.00 (0.01–0.01)	
Kynurenine to tryptophan ratio *	B	0.09 (0.07–0.10)	0.08 (0.07–0.09)	0.117	0.00 (−0.02–0.01)	0.156
	S	0.08 (0.07–0.09)	0.08 (0.07–0.09)	0.955	0.00 (−0.01–0.01)	

Data are mean ± SD or median (interquartile range). * log-transformed variables are used for the analyses. B, branched-chain amino acids (BCAA) group; eGFR, estimated glomerular filtration rate; HDL-C, high-density lipoprotein cholesterol; HOMA-IR, homeostasis model assessment of insulin resistance; LDL-C, low-density lipoprotein cholesterol; MMSE, Mini Mental State Examination; QIDS, Quick Inventory of Depressive Symptomatology; S, Soy protein group; SAS, Starkstein Apathy Scale.

**Table 5 nutrients-14-03917-t005:** Changes in serum levels of insulin-like growth factor 1, follistatin, and brain-derived neurotropic factor during the intervention.

	Group	Baseline	24 Weeks	*p* for Time	Changes	*p* for Groups
IGF-1 (mg/mL)	B	102 ± 32	99 ± 32	0.292	−3 ± 13	0.316
	S	101 ± 40	105 ± 33	0.589	4 ± 28	
Follistatin (pg/mL) *	B	1312 (711–1807)	1419 (868–1857)	0.675	68 (−316–319)	0.611
	S	1091 (889–2106)	1232 (907–2225)	0.595	−38 (−149–366)	
BDNF (pg/mL)	B	23,667 ± 4452	23,011 ± 5020	0.511	−655 ± 4376	0.439
	S	22,130 ± 4919	20,093 ± 6210	0.214	−2037 ± 6068	

Data are mean ± SD or median (interquartile range). * log-transformed variables are used.

## Data Availability

Data for this manuscript can be obtained by contacting the corresponding author with a reasonable request.

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
