# Peer review of "Effects of Branched-Chain Amino Acids on Skeletal Muscle, Glycemic Control, and Neuropsychological Performance in Elderly Persons with Type 2 Diabetes Mellitus: An Exploratory Randomized Controlled Trial"

_nutrients, 2022, doi:10.3390/nu14193917_

Round 1

Reviewer 1 Report

This paper examines how a BCAA intervention improves muscle characteristics, glycemic control and tryptophane metabolism. 

My major comment, which will modify the outcomes of this paper is that the authors do not mention the exercise intervention until we reach the methods section. I was certain that this paper focused only on BCAA supplementation. So, I was surprised to read on page 3, lines 131 to 137 that aerobic and resistance exercise was included in this study.

So, according to this, it seems possible that all of the results obtained in this study may be caused by this increase in physical activity. It is already known that T2D older adults have decreased PA levels. As such, increasing this may have influenced the results. Without a proper control arm, we cannot conclude that BCAA and soy supplementation increased muscle strength. 

BCAA supplementation levels varies across studies. Why the authors used 8g?

The authors mention that the recruitment of participants did not go as planned because of the COVID-19 pandemic. However, they have reached an adequate sample based on their sample size calculation.

They authors mention "tryptophan metabolism". Is this the correct term? They've only looked at tryptophan levels. 

The secondary objectives of this study are a bit confusing. In the title of the study, 3 topics are covered: muscle, glycemic control and tryptophan metabolism. However, in the outcomes section in the methods, there is a very long list of variables that were never mentioned in the introduction. The authors should clean this section and make sure it reflects what they want to examine in this study.

Author Response

The authors appreciate the care with which you reviewed our paper. Your comments were very helpful in improving this manuscript. If there are any further misunderstandings or our interpretations are inadequate, we will be happy to make further corrections or additions if you could point out any particular issues that remain. Your comments are highlighted in black bold. We are responding point-by-point as follows.

  1. My major comment, which will modify the outcomes of this paper is that the authors do not mention the exercise intervention until we reach the methods section. I was certain that this paper focused only on BCAA supplementation. So, I was surprised to read on page 3, lines 131 to 137 that aerobic and resistance exercise was included in this study. So, according to this, it seems possible that all of the results obtained in this study may be caused by this increase in physical activity. It is already known that T2D older adults have decreased PA levels. As such, increasing this may have influenced the results. Without a proper control arm, we cannot conclude that BCAA and soy supplementation increased muscle strength.

Response 1

Thank you for your important remarks. As you pointed out, it is possible that exercise affected the results. Therefore, we discussed such possibilities in the Discussion (p.15 lines 494-499). Resistance training increases skeletal muscle mass and/or strength in elderly persons with T2D, but a systematic review showed that protein supplementation combined with resistance training was more effective compared with resistance training alone. Furthermore, studies conducted by Yamamoto et al. and Memelink et al. demonstrated that the increases in skeletal muscle mass and/or strength with the combination of supplementation with amino acids or protein and resistance training were more pronounced compared with resistance training alone. Therefore, it is possible that the increases in muscle strength with BCAA or soy protein supplementation observed in this study were not due only to resistance training. Those points were added to the Discussion (p.13 lines 377-385). In addition, we revised the Introduction to indicate that in our study the supplementation of amino acids was accompanied by standard exercise based on recommendations of the American Diabetes Association (p.2 lines 94-95).

  1. BCAA supplementation levels varies across studies. Why the authors used 8g?

Response 2

Thank you for your question. A study examining muscle protein synthesis (MPS) in elderly men showed that supplementation with 0.052 g/kg leucine, 0.0116 g/kg isoleucine, and 0.0068 g/kg valine significantly increased MPS compared with controls (1). For a body weight of 60 kg, this corresponds to a dose of 3 g leucine, 0.7 g isoleucine, and 0.4 g valine. One package of the used supplement contained 4 g of BCAA (4 g of leucine, 2 g of isoleucine, and 2 g of valine). Since we considered that 8 g of BCAA could stimulate MPS, we used two packets of the supplement. We have revised Materials and Methods (p.3 lines 122-128).

  1. Rieu I, Balage M, Sornet C, et al. Leucine supplementation improves muscle protein synthesis in elderly men independently of hyperaminoacidaemia. J Physiol. 2006;575(Pt 1):305-15.

  1. The authors mention that the recruitment of participants did not go as planned because of the COVID-19 pandemic. However, they have reached an adequate sample based on their sample size calculation.

Response 3

We apologize for the incomplete explanation. The sample size calculations indicated that 16 participants were required per group. However, we set the sample size at 20 participants per group to account for dropouts. In fact, we were able to recruit the expected number in the BCAA group, but we could only recruit 18 participants in the soy protein group, with two dropping out and one accidentally given BCAA. This reduced the number of participants who actually received soy protein to 15.

  1. They authors mention "tryptophan metabolism". Is this the correct term? They've only looked at tryptophan levels.

Response 4

Thank you for your question. As you pointed out, we only examined the serum levels of tryptophan or some of its metabolites including kynurenine or kynurenic acid but not all of them. In addition, clinically, changes in neuropsychological functions are more important than those in tryptophan metabolites. We revised the Discussion (p.12 lines 340-341, p.14 line 441)

  1. The secondary objectives of this study are a bit confusing. In the title of the study, 3 topics are covered: muscle, glycemic control and tryptophan metabolism. However, in the outcomes section in the methods, there is a very long list of variables that were never mentioned in the introduction. The authors should clean this section and make sure it reflects what they want to examine in this study.

Response 5

Thank you for your comment. We have removed body weight and body composition which were not presented in Results. We have not removed other secondary outcomes because they were used to assess the safety of BCAA supplementation (p.4 lines 149-158).

Additional comment

  1. We have made some changes to the manuscript for more accurate and consistent descriptions (p.2 line 57 and line 85, p.6 line 258, and p.15 line 465 and line 471).

Reviewer 2 Report

The topic of the review is interesting and relevant. However, there are some areas that can be improved. Here are some suggestions.

1-The title and objective of the study mention 3 outcomes, which are skeletal, glucose, and trytophan metabolism. However, it is not consistent throughout the writing as some area discusses the outcome of depressive symptoms Please kindly rewrite the title or focus on the outcomes that are significant in this study.

2-Please separate the conclusion from the limitation.

3-Add on the strength of this study.

4-How do the authors ensure the participants follow the instruction to perform aerobic exercise and comprehensive resistance training at least three times a week?

5-Lines 496-Authors should discuss the results and how they can be interpreted from the perspective of previous studies and of the working hypotheses. The findings and their implications should be discussed in the broadest context possible. Please kindly double-check these sentences.

Thank you.

Author Response

The authors appreciate the care with which you reviewed our paper. Your comments were very helpful in improving this manuscript. If there are any further misunderstandings or our interpretations are inadequate, we will be happy to make further corrections or additions if you could point out any particular issues that remain. Your comments are highlighted in black bold. We are responding point-by-point as follows.

  1. The title and objective of the study mention 3 outcomes, which are skeletal, glucose, and trytophan metabolism. However, it is not consistent throughout the writing as some area discusses the outcome of depressive symptoms Please kindly rewrite the title or focus on the outcomes that are significant in this study.

Response 1

Thank you for your comments. We revised the title eliminating mention of tryptophan metabolism and added “neuropsychological functions”. We feel this is reasonable as we assessed not only depressive symptoms but also global cognition (MMSE) and motivation (SAS) (p.1 line 3).

  1. Please separate the conclusion from the limitation.

Response 2

Thank you. We newly added a separate Conclusion (p.16 lines 516-524).

  1. Add on the strength of this study.

Response 3

We added the strengths of the current study in the Discussion (p.15 lines 486-493) We changed the previous heading, which was “Limitations” to “Strengths and limitations”.

  1. How do the authors ensure the participants follow the instruction to perform aerobic exercise and comprehensive resistance training at least three times a week?

Response 4

Although the clinical research coordinator regularly telephoned or e-mailed the participants to check on exercise status, the only objective data were physical activity levels measured by an accelerometer. We have moved the sentence “A clinical research coordinator (CRC) monitored supplement intake and exercise once a week using e-mail or telephone to assure compliance with and to check for adverse effects” to the third paragraph because “the intervention” included both BCAA or soy protein supplementation and exercise (p.3 lines 141-143). The newly added sentence emphasizes that the CRC checked for compliance with exercise as well as supplement intake.

  1. Lines 496-Authors should discuss the results and how they can be interpreted from the perspective of previous studies and of the working hypotheses. The findings and their implications should be discussed in the broadest context possible. Please kindly double-check these sentences.

Response 5

Thank you for pointing this out. We apologize for not deleting the sentences referred to. This was unintentional. We have now deleted those sentences.

Additional comment

We have made some changes to the manuscript for more accurate and consistent descriptions (p.2 line 57 and line 85, p.6 line 258, and p.15 line 465 and line 471).

Round 2

Reviewer 1 Report

The authors have adequately replied to my comments. It is now suitable for publication.